# Design of an Herbal Preparation Composed by a Combination of *Ruscus aculeatus* L. and *Vitis vinifera* L. Extracts, Magnolol and Diosmetin to Address Chronic Venous Diseases through an Anti-Inflammatory Effect and AP-1 Modulation

**DOI:** 10.3390/plants12051051

**Published:** 2023-02-26

**Authors:** Raffaella Nocera, Daniela Eletto, Valentina Santoro, Valentina Parisi, Maria Laura Bellone, Marcello Izzo, Alessandra Tosco, Fabrizio Dal Piaz, Giuliana Donadio, Nunziatina De Tommasi

**Affiliations:** 1Ph.D. Program in Drug Discovery & Development, Università degli Studi di Salerno, 84084 Fisciano, Italy; 2Department of Pharmacy, Università degli Studi di Salerno, 84084 Fisciano, Italy; 3Department of Medicine and Surgery, Univesità degli Studi di Ferrara, 44121 Ferrara, Italy; 4Department of Medicine, Surgery and Dentistry, Università degli Studi di Salerno, 84081 Baronissi, Italy

**Keywords:** *Ruscus aculeatus* L., *Vitis vinifera* L., magnolol, diosmetin, anti-inflammatory effects, transcriptional factors, AP-1, MCP-1, venous disease

## Abstract

Chronic venous disease (CVD) is an often underestimated inflammatory pathological condition that can have a serious impact on quality of life. Many therapies have been proposed to deal with CVD, but unfortunately the symptoms recur with increasing frequency and intensity as soon as treatments are stopped. Previous studies have shown that the common inflammatory transcription factor AP-1 (activator protein-1) and nuclear factor kappa-activated B-cell light chain enhancer (NF-kB) play key roles in the initiation and progression of this vascular dysfunction. The aim of this research was to develop a herbal product that acts simultaneously on different aspects of CVD-related inflammation. Based on the evidence that several natural components of plant origin are used to treat venous insufficiency and that magnolol has been suggested as a putative modulator of AP-1, two herbal preparations based on Ruscus aculeatus root extracts, and Vitis vinifera seed extracts, as well as diosmetin and magnolol, were established. A preliminary MTT-based evaluation of the possible cytotoxic effects of these preparations led to the selection of one of them, named DMRV-2, for further investigation. First, the anti-inflammatory efficacy of DMRV-2 was demonstrated by monitoring its ability to reduce cytokine secretion from endothelial cells subjected to LPS-induced inflammation. Furthermore, using a real-time PCR-based protocol, the effect of DMRV-2 on AP-1 expression and activity was also evaluated; the results obtained demonstrated that the incubation of the endothelial cells with this preparation almost completely nullified the effects exerted by the treatment with LPS on AP-1. Similar results were also obtained for NF-kB, whose activation was evaluated by monitoring its distribution between the cytosol and the nucleus of endothelial cells after the different treatments.

## 1. Introduction

Chronic venous disease (CVD) of the lower limbs represents a widespread vascular pathology, with a prevalence in the population of about 20–25% [1,2,3]. This pathology is generated by an abnormal increase in pressure in veins, resulting in morphological changes that significantly compromise their functions [4]. The main clinical sign of CVD is varicose veins, mainly involving lower limbs and consisting of an evident thickening of vein walls and in changes in their composition [5]. Further disorders related to CVD are telangiectasias and reticular veins, and if no therapeutic treatment is conducted, leg heaviness and pain result [6] (Figure 1). CVD is related to several factors such as age, cardiovascular diseases, postural factors, overweight, smoking, constrictive clothing, periods of immobility, constipation, fiber intake, and hormonal aspects [4]. Its etiology complex and multifactorial; numerous genetic factors underlying this pathology have been suggested, and many single nucleotide polymorphisms or other mutations seem to be related to CVD occurrence and severity [7]. However, epigenetic aspects are emerging as those playing a key role in the onset of this dysfunction. In particular, the common inflammatory transcription factor AP-1 (activator protein-1) activation has been shown to be involved in worsening of CVD symptoms and in the remodeling of veins towards the formation of varicose veins [8].

AP-1 cooperates with NF-kB playing an important pro-inflammatory role. AP-1 is, in fact, a family of proteins that bind specific DNA sequences termed the 12-*o*-tetradecanoyl-phorbol 13-acetate (TPA) response element (5′-TGAG/CTCA-3′) present in the promoter of genes implicated in cell proliferation, differentiation, stress response and inflammation [9].

Structurally, AP-1 is a dimer obtained by the homo or heterodimerization of members of the c-Fos (c-Fos, FosB, Fra-1 and Fra-2) and c-Jun (c-Jun, c-JunB and c-JunD) families. The induction and activation of AP-1 rely mainly on MAP kinases, including ERK, and occur in response to growth factors, cytokines and various stress in cultured cells. Additionally, AP-1 is implicated with NF-kB in the induction of interleukins and monocyte-chemoattractant protein-1 (MCP-1) genes. MCP-1 is a potent chemokine involved in leukocytes recruitment from blood to vein walls and is suggested to be responsible for severe vein remodeling [10]. Consequently, AP-1 inhibition has been proposed as an appealing target to prevent the progression of CVD [11,12].

Although a number of pharmacological approaches against CVD have been proposed, recurrent symptoms with an increasing frequency in the years following the end of treatment have not been solved [13]. Possible reasons can be traced back to approaches focused on secondary endpoints of the disease, such as symptom control of CVD through vessel strengthening and/or inflammation care, without addressing its cause. Indeed, many natural components from plant sources are used to treat venous insufficiency and varicose vein [14,15,16,17]. However, these products are used for their antioxidant and anti-inflammatory effects. They primarily protect endothelial tissue from oxidative damage, reducing many CVD-associated disorders, but producing a negligible effect on the pathology progression [18].

Several natural compounds and/or extracts have been shown to significantly reduce inflammation and protect the veins from different kinds of insults. Magnolol, a neolignan obtained from the bark of *Magnolia officinalis* (Rehder & E. Wilson) and other species of the Magnoliaceae family, have shown a broad spectrum of bio-activities. It was reported previously to affect vascular remodeling processes associated with neointima formation and atherosclerosis and has been reviewed for its anti-aging and protective effects on the skin [19,20]. Recently, several of these activities were related to the ability of magnolol to inhibit AP-1 signaling [21,22]. Diosmetin is an *o*-methylated flavone, the aglycone part of diosmin that occurs naturally in citrus fruits. Pharmacologically, it exhibits anticancer, antimicrobial, antioxidant, oestrogenic and anti-inflamatory activities. In Europe, diosmin and diosmetin have been widely used for a long time as phlebotonics and vascular protectors in different types of pharmaceutical and nutritional products [23,24]. *Ruscus aculeatus* L. (Asparagaceae) roots hydroalcoholic extract, which contains ruskogenin, neuroruskogenin and other saponins, has been used for many years to decrease sensations of leg heaviness and leg swelling. Among its confirmed activities, one of the major effects of this extract is the vasoconstrictive activity related to α-1 and α-2 receptor agonism in the vessel wall and the release of norepinephrine from adrenergic nerve endings [25]. *Vitis vinifera* L. (Vitaceae) seeds extract, which is rich in polyphenols and oligomeric proanthocyanidins [26,27], has been used for cardiovascular disease prevention, venous dysfunction management, and type 2 diabetes treatment. The efficacy of this extract was proved by several clinical studies [28,29,30].

The aim of this research was to develop an herbal product composed of a mixture of magnolol, diosmetin, *R. aculeatus* root and *V. vinifera* seeds extracts, and to evaluate its efficacy in reducing CVD-related inflammation and blocking the hyper-activation of AP-1. To this aim, we prepared two mixtures, named DMRV-1 and 2, and tested their toxicity towards human endothelial cells (HUVECs); the preparation providing the best results was then assayed for its anti-inflammatory activity and effects on AP-1 expression and activation in the same cell model.

## 2. Results and Discussion

### 2.1. Extracts Characterization

*R. aculeatus* roots are rich in steroidal saponins, which can be considered markers of this genus, and show several biological activities involving the venous system, such as the vasoconstrictive and venotonic effects exerted by ruscogenin and neoruscogenin [31]. *V. vinifera* seeds are rich in procyanidins, which are widely used in the treatment of circulatory disorders and inflammation [28]. Therefore, an LC-MS/MS-based quantification analysis was performed on ruscogenin for the *R. aculeatus* extracts and procyanidin B1 for the *V. vinifera* seeds extracts, as representatives of steroidal saponins and procyanidins, respectively (Appendix A). Our results demonstrated that *R. aculeatus* root extract contains 97 ± 1 ppm (*w*/*w*) ruscogenin and *V. vinifera* seed extract contains 0.76 ± 0.03% (*w*/*w*) procyanidin B1. These quantities were consistent with those expected for these extracts [32,33].

### 2.2. Cytotoxicity Assays

The potential in vitro cytotoxicity of herbal preparation constituents (magnolol, diosmetin, *V. vinifera* and *R. aculeatus* extracts) was evaluated on human endothelial cells (HUVECs) using the MTT assay. This step allowed us to assess the safety of these preparations, and also to determine the optimal conditions to perform the subsequent experiments. HUVECs were, therefore, exposed to increased concentrations of the four components and cell viability was evaluated at 24 h (Figure 2). Half maximal inhibitory concentrations (IC_50_) of 7.2 μg/mL, 6.2 μg/mL, 42 μg/mL and 58 μg/mL were measured for magnolol, diosmetin, and the *R. aculeatus* and *V. vinifera* extracts, respectively.

The results obtained demonstrated that diosmetin was toxic towards HUVECs, thus suggesting the inclusion of low concentrations of this compound in the formulation. Moreover, a non-negligible and dose-dependent cytotoxicity was measured also for the other components. Data obtained for *V. vinifera* seeds extract were in agreement with those previously reported [34], while to the best of our knowledge, it is the first time that toxicity towards HUVECs has been recorded for diosmetin and *R. aculeatus* root extract. Cytotoxicity of magnolol against endothelial cells was not previously observed, but this compound was only evaluated at low concentrations (≤2.7 μg/mL) [35].

Based on these results and on those previously published relating to the use of these compounds in the treatment of venous pathologies [19,20,21,22,30,31], two herbal mixtures (DMVR-1 and DMVR-2) differing in the relative amounts of the four components were prepared. The cytotoxicity of these two formulations against HUVECs was assayed by MTT, by exposing the cells to 25 μg/mL and 40 μg/mL of each preparation for 24 h (Figure 3). Although DMRV-1 showed toxicity, the decrease in cell viability observed following treatment with the highest tested concentration of DMRV-2 was such that it did not bias subsequent analyses. Therefore, this herbal preparation was selected for subsequent studies and evaluations.

### 2.3. Anti-Inflammatory Activity

DMRV-2 was tested on HUVECs pre-treated with bacterial lipopolysaccharides (LPS) to evaluate the ability of the preparation to revert or, at least, significantly reduce, the inflammatory effects induced by LPS. Incubation of HUVECs with 0.1 µg/mL LPS for 4 h produced a massive secretion of IL-6 and IL-8, which persisted for at least 6 h once the LPS was removed from the culture medium (Figure 4). When these inflamed HUVECs were exposed to a non-toxic concentration (25 µg/mL) of DMRV-2 for 3 h or 6 h, the secretion of IL-6 and IL-8 was significantly (*p* < 0.05) reduced, becoming similar to that observed for the non-stimulated cells. These data suggest that DMRV-2 exerts an efficient anti-inflammatory effect after 3 h of treatment, lasting at least for other 3 h.

The ability of DMRV-2 to lower the secretion of interleukins 6 and 8 is not only a marker of a significant reduction in inflammation, but also has a direct beneficial effect on vascular remodeling [36]. Indeed, it has been demonstrated that interleukins and other cytokines regulate the balance between different collagen types in venous and arterial tissues [37] as well as the structure and stability of elastin [38], thus significantly affecting the stiffness of blood vessels and capillaries, especially those not covered by smooth muscles.

### 2.4. Modulation of NF-κB and AP-1

Nuclear factor kappa-light-chain-enhancer of activated B cells (NF-κB) and activator protein 1 (AP-1) are the main transcription factors that orchestrate the expression of many genes involved in inflammation [39]. Several studies suggest that NF-κB and AP-1 are regulated by the same intracellular signal transduction cascades and are activated simultaneously by the same multitude of stimuli [40,41]. NF-κB is activated by various extracellular inflammatory signals (i.e., TNFα, IL-1, and CD40 ligand), inducing its migration from the cytosol to the nucleus, where it modulates the expression of proteins variously involved in the cellular response to inflammation [38]. On the other hand, AP-1 is induced by the activation of the extracellular signal-regulated kinase (ERK) subgroup of mitogen-activated protein kinases (MAPKs) [42,43,44].

The pivotal and interdependent role played by NF-κB and AP-1 in the inflammatory processes prompted us to investigate the effect of DMRV-2 on the activity of these two proteins. In cells subjected to inflammation, NF-κB moves from the cytosol to the nucleus; therefore, we measured the concentration of this protein in the two cell compartments following incubation of the HUVECs with LPS and after 3 h or 6 h treatment with DMRV-2. Densitometric analysis of the resulting WB showed that LPS-induced inflammation produced a doubling in the concentration of NF-κB in the HUVEC nucleus (*p* < 0.05). The 3-h and 6-h DMRV-2 treatments of inflamed HUVECs (Figure 5 and Appendix A) significantly reduced this increment (*p* < 0.05), but they were unable to restore the protein level measured in non-inflamed cells.

The opposite trend was observed for the protein concentration in the cytosol. DMRV-2 was, therefore, capable of inhibiting the activation of NF-κB induced by LPS treatment, even if in our experimental conditions this inhibition seemed to be only partial.

Real-time quantitative PCR was used to evaluate the modulation of AP-1 expression and activity by DMRV-2. Structurally, AP-1 is a heterodimer composed of different combinations of two proteins belonging to the Jun and Fos families; the predominant form in most cells is the Fos/Jun heterodimers, while a Jun/Jun homodimer has been observed less frequently [45,46]. Functionally, AP-1 regulates the expression of many genes involved in several pathological and physiological functions; among them, one of the most relevant in the inflammatory process is that encoding for the monocyte-chemoattractant protein (MCP)-1, whose expression is strongly enhanced by the activation of AP-1 [47]. Therefore, we monitored the transcription of c-Jun, c-Fos and mcp-1 genes in HUVECs following the LPS-induced inflammation and the subsequent 3- or 6-h incubation with DMRV-2 (Figure 6). Incubation of HUVECs with LPS produced a strong and persistent stimulation of the transcription of both mcp-1 and c-jun genes. Indeed, the levels of mRNAs of these two genes were significantly (*p* < 0.05) increased by a 4-h LPS treatment, reached their maximum 3 h after LPS removal from the medium, and subsequently started to decrease. Interestingly, incubation of LPS-treated HUVECs with DRMV-2 for 3 or 6 h significantly (*p* < 0.05) reduced mcp-1 transcription (Figure 6A). This effect could be due to a drastic inhibition of the transcription of the c-jun gene, as indicated by the observation that the levels of the relative mRNA after the incubation with DMRV-2 were even lower than those observed in the uninflamed HUVECs (Figure 6B). Conversely, no significant effects on c-fos transcription were detected following the LPS-induced inflammation or the incubation with the preparation (Figure 6C).

Given the particular nature of AP-1, which can exist as a Jun/Fos heterodimer or a Jun/Jun homodimer, these data suggest that the inflammation induced by LPS could favor the homodimer formation. Accordingly, when treating the inflamed cells with a magnolol-containing preparation, the c-Jun mRNA returned to a basal level, thus restoring the correct balance with c-Fos and favoring the normal formation of the heterodimer.

## 3. Materials and Methods

### 3.1. Reagents and Materials

Ultra-pure acetonitrile, water, methanol, and formic acid for LC-MS analysis were purchased from Romil Ltd. Pure Chemistry (Cambridge, UK). The solvents for the extraction were purchased from Sigma Chemicals Company (Milan, Italy). For the quantitative analyses, the following standards were used: ruscogenin and procyanidin B1 obtained from DBA Italia s.r.l. (Milan, Italy), magnolol obtained from Activate Scientific GmbH (Am Mitterweg 12, 83209, Prien am Chiemsee, Germany), and diosmetin obtained from Nutradade s.r.l. (Milan, Italy). Anti-Erk2 (mouse monoclonal sc-1647) was obtained from Santa Cruz Biotechnology (Delaware, CA, USA); and anti-GAPDH (mouse monoclonal 437000) and phospho-p44/42 MAPK (Erk1/2) (Thr202/Tyr204) (rabbit monoclonal #4376) were obtained from Cell Signaling (Danvers, MA, USA). The appropriate peroxidase-conjugated secondary antibodies were from Jackson Immuno Research (Baltimore, PA, USA). *V. vinifera* seeds aqueous extract was provided by Nutradade s.r.l. *R. aculeatus* L. (Asparagaceae) roots were collected in 2021 in the rural area of Salerno (Italy) and identified by Prof. Vincenzo De Feo (University of Salerno). A voucher specimen (DF212/2022) was deposited at the University of Salerno.

### 3.2. Extracts and Formulations Preparation

A 350 g sample of *R. aculeatus* dried and powdered roots were extracted with EtOH-H2O 1:1 at a ratio of 1:5 (*w*/*v*). The extraction was performed using a 320 W Ultrasonic bath (Branson 2510E-MTH, Bransonic^®^, Milano, Italy) for 15 min. The extract, after filtration, was dried under vacuum, frozen, lyophilized to remove the excess water and stored at 4 °C for further analysis; the yield was 10.5%.

The two herbal preparations were obtained by directly mixing lyophilized power of the four components. DMRV-1: diosmetin 6% (*w*/*w*), magnolol 10% (*w*/*w*), *V. vinifera* seeds extract 48% (*w*/*w*), and *R. aculeatus* extract 36% (*w*/*w*). DMRV-2: diosmetin 11% (*w*/*w*), magnolol 9% (*w*/*w*), *V. vinifera* seeds extract 45% (*w*/*w*), and *R. aculeatus* 35% (*w*/*w*).

### 3.3. Quantitative Analysis of Ruscogenin and Procyanidin B1

An MRM method was used to quantify ruscogenin and procyanidin B1 in the *R. aculeatus* root and *V. vinifera* seed extracts, respectively. The analyses were carried out using an API6500 Q-Trap (ABSciex Foster City, CA, USA) coupled with an A NexeraX2 UHPLC apparatus (Shimadzu, USA), operating in positive ion mode. Instrumental parameters were optimized using pure compounds. A Luna^®^ Omega 100 mm × 1.6 mm, 3 µm (100 Å) column (Phenomenex^®^, Castel Maggiore, Bologna, Italy) was used for chromatographic separation with water acidified by 0.1% formic acid *v/v* (solvent A) and acetonitrile (solvent B) as eluent. The gradient used was a first linear step from 5% to 25% of acetonitrile (eluent B) in 10 min followed by another faster gradient step up to 100% of B in 8 min. The flow rate was set to 0.3 mL/min and the injection volume was 10 µL for standards and samples. The method was validated and was accurate and linear over a concentration range from 20 to 750 ng/mL for both compounds.

### 3.4. Cell Culture

The HUVEC/TERT2 cell line was purchased from the American Type Culture Collection (ATCC No. CRL-4053). The cells were grown in vascular cell basal medium supplemented with endothelial cell growth-Kit-VEGF containing hydrocortisone, rhFGF, rhVEGF, rh-IGF-1, ascorbic acid, Heparin sulphate, FBS, and hEGF, L-glutamine (LGC, standard Charles City, IA), at 37 °C in a humidified atmosphere of 5% CO_2_. To ensure logarithmic growth, cells were subcultured every 3 days. Stock solutions of compounds in DMSO were stored in the dark at 4 °C. Appropriate dilutions were prepared in the culture medium immediately before use. In all experiments, the final concentration of DMSO did not exceed 0.1% (*v*/*v*).

### 3.5. Cell Viability Assay

Human Umbilical Vein Endothelial Cells were plated in 96-well plates at a cell density of 1*10^5^ cells/well. After 24 h, the cell lines were incubated for 24 h in the presence of magnolol (1.3 to 10 µg/mL), diosmetin (3.1 to 25 µg/mL), *R. acuelatus* extract (6.3 to 50 µg/mL), *V. vinifera* extract (6.3 to 50 µg/mL), and of the herbal preparation at final concentrations in the range 2.5–0 μg/mL of magnolol. The number of viable cells was quantified by the MTT [3-(4,5-dimethylthiazol-2-yl)-2,5-diphenyl tetrazolium bromide] assay. Absorption at 550 nm for each well was assessed using Multiskan GO (Thermo Scientific, Waltham, MS, USA). The viability % was obtained using the following formula:% Viability = (mean OD treated well)/(mean OD control well)∗100

Half-maximal inhibitory concentration (IC_50_) values were calculated using GraphPad Prism 8. The experiments were performed in technical triplicates.

### 3.6. Competitive ELISA Assays

Enzyme-linked immunosorbent assay (ELISA) kits for IL-6 and IL-8 were used from Diaclone Company (France). The HUVECs were plated in 6-well plates at a cell density of 3 × 10^5^ cells/well. The HUVECs were incubated for 4 h with LPS (0.1 µg/mL), and then incubated for 3 h and 6 h with DMVR-2 (2.5 µg/mL of magnolol). The supernatants were centrifuged according to the protocol and stored at −20 °C until testing. The procedures for measuring IL-6 and IL-8 levels followed the instructions provided by the kit manufacturers. Each point was tested in triplicate and each experiment was repeated twice.

### 3.7. NF-κB Translocation Analysis

HUVECs were plated in 60-mm culture dishes (6 × 10^5^ cells/dish), incubated with LPS (0.1 µg/mL) for 4 h, and then incubated with DMVR-2 (25 µg/mL) for 3 h and 6 h and the respective controls. The cells were treated with trypsin and washed twice with PBS. Subsequently, the cell pellet was subjected to cell lysis and separation of the cytoplasmic and nuclear protein extraction using a NE-PER Nuclear and Cytoplasmic Extraction Kit (Thermo Scientific). The protein concentration was determined by a Bradford-Solution for Protein Determination (Applichem, Monza, Italy) using BSA as a standard. The proteins were fractionated on SDS-PAGE and transferred into nitrocellulose membranes. After electro transferring, the nitrocellulose membrane was blocked with 10% nonfat dry milk for 1 h at room temperature and then immunoblotted with appropriate primary antibodies, against Nf-kB p65 (F-6) (Santa Cruz Biotechnology), GAPDH (Santa Cruz Biotechnology) and Nucleolin (Sigma-Aldrich), O/N at 4 °C. The signals were visualized with the appropriate horseradish peroxidase-conjugated secondary antibody and enhanced chemiluminescence (Amersham Biosciences-GE Healthcare, NY, USA). Densitometric analyses were carried out using the ImageJ software Java8. The experiments were performed in duplicate. 

### 3.8. RNA Isolation and Quantitative Real-Time-PCR (qRT-PCR)

The HUVECs were plated in 60-mm culture dishes at a cell density of 6 × 10^5^ cells/dish. They were first incubated for 4 h with LPS (0.1 µg/mL), and then incubated for 3 h or 6 h with DMVR-2 (2.5 µg/mL of magnolol). Total RNA was isolated from cultured cells using Trizol Reagent (Life Technologies, Grand Island, NY, USA) according to the manufacturer’s instructions, spectrophotometrically quantified, and the quality assessed by agarose gel electrophoresis. Then, 700 ng of each RNA was retrotranscribed by M-MLV Reverse Transcriptase (GeneSpin S.r.l, #STS-MRT, Italy), and Real-Time PCR was performed with Light-Cycler^®^ 480 (Roche Diagnostics GmbH, Mannheim, Germany) using SYBR Green I Master Mix (Life Technologies). The forward and reverse primers were used at a concentration of 10 µM. The PCR program included an initial cycle at 95 °C for 5 min, followed by 45 cycles of denaturation at 95 °C for 10 sec, annealing at 60 °C for 20 sec, and extension at 72 °C for 15 sec. HPRT1 was used as a housekeeping gene. The following primer sets were used for Real-Time PCR to assay specific mRNAs (Fw hCCL2 -AGC AAG TGT CCC AAA GAA GC-″-; Rev hCCL2 ″-CCT GAA CCC ACT TCT GCT TGG-3′; Fw hFOS ″-CAG ACT ACG AGG CGT CAT CC-″; Rev hFOS ″-TCT GCG GGT GAG TGG TAG TA-″; Fw hJUN ″-TGA GTG ACC GCG ACT TTT CA-″; Rev hJUN ″-TTT CTC TAA GAG CGC ACG CA-″. The relative gene expression was compared using the 2^−ΔΔCt^ method as described elsewhere [48]. A value of significance was established when *p* < 0.05 and variances are displayed as standard errors. All statistical analyses were performed using GraphPad Prism 8.

### 3.9. Statistical Analysis

Data are reported as the mean values ± SD from at least three experiments performed in duplicate (n ≥ 6), showing similar results. Differences between treatment groups were analyzed by Student ‘s *t*-test. Differences were considered significant when *p* < 0.05.

## 4. Conclusions

We used a mixture of plant extracts whose anti-inflammatory activity was well documented and magnolol, a plant metabolite supposed to inhibit AP-1 activity [22], to create an herbal preparation putatively effective against CVD-related inflammation. The efficacy of this preparation, named DMRV-2, was tested on a model of human endothelial cells subjected to a severe inflammatory stimulus. The obtained results demonstrated the capability of this preparation to counteract cytokine release and NF-κB and AP-1 inflammation-induced activation. These results suggest the need to carry out further studies aimed at evaluating the use of this preparation as a possible support or in combined approach for the treatment of CVD. It is, however, important to underline the cytotoxicity at high concentrations observed for some of the DMRV-2 components, although data reported in previous studies demonstrated that the use in vivo of these substances/extracts is quite safe [49,50,51,52]. The data presented here suggest that this herbal preparation of well-known botanicals at a specific ratio could produce a mixture with a much wider spectrum of action to interact with multiple pathways involved in CVD onset and progression. Moreover, interesting information on the effects of anti-inflammatory compounds on AP-1 modulation emerged from this study. Indeed, our data allow hypothesizing a regulatory mechanism of this transcriptional factor based on the restoration of the correct stoichiometric ratio between the c-Jun and c-Fos proteins, which determine a change in the equilibrium between the homo-dimeric and hetero-dimeric forms of AP-1. Hence, we suggest further mechanism-based elucidation of this mixture for CVD management in clinical studies.

## Figures and Tables

**Figure 1 plants-12-01051-f001:**
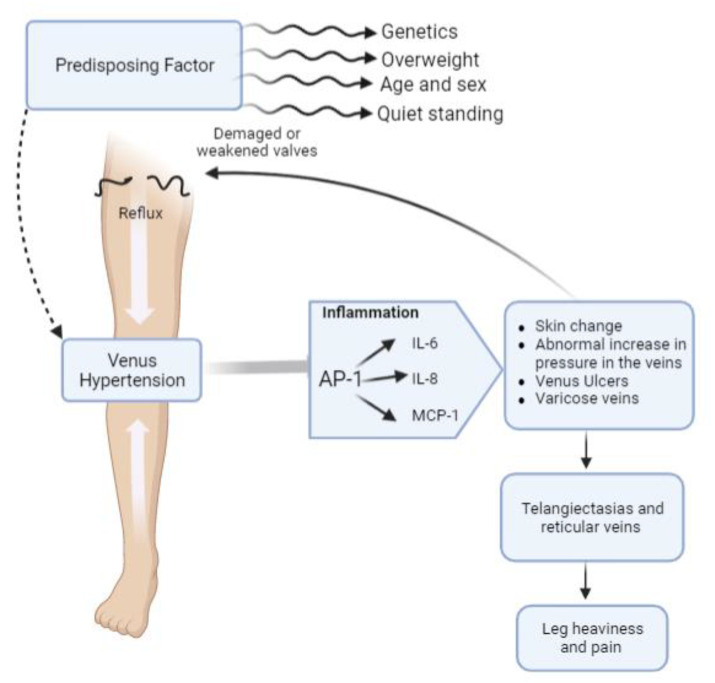
Schematic representation of chronic venous disease (CVD) pathogenesis and major clinical signs.

**Figure 2 plants-12-01051-f002:**
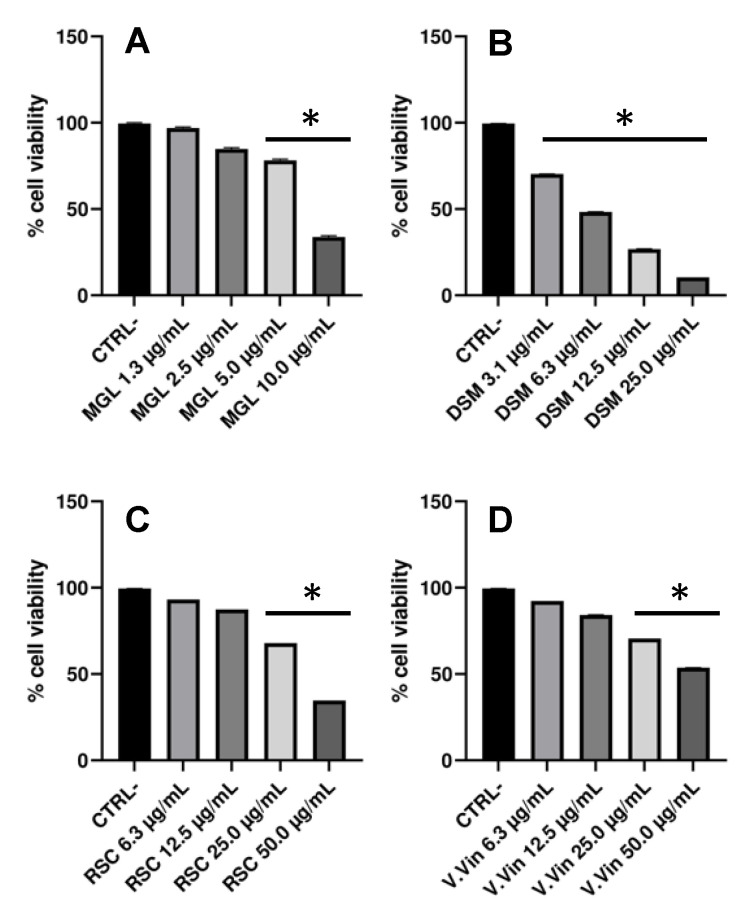
HUVECs viability measured by MTT assay (Mean ± SEM of 6 independent measurements) following different treatments with each compound and extract selected: the viability of HUVEC cells after a 24 h incubation with different concentrations of magnolol (MGL) (1.3, 2.5, 5.0, 10.0 μg/mL) (**A**); diosmetin (DSM) (3.1, 6.3, 12.5, 25.0 μg/mL) (**B**); *R. aculeatus* root extract (RSC) (6.3, 12.5, 25.0, 50.0 μg/mL) (**C**); and *V. vinifera* seeds extract (V.Vin) (6.3, 12.5, 25.0, 50.0 μg/mL) (**D**). In all the panels, CTRL- represents the negative control, consisting of HUVECs incubated with the cell medium for 24 h. * *p* < 0.05 vs. CTRL-.

**Figure 3 plants-12-01051-f003:**
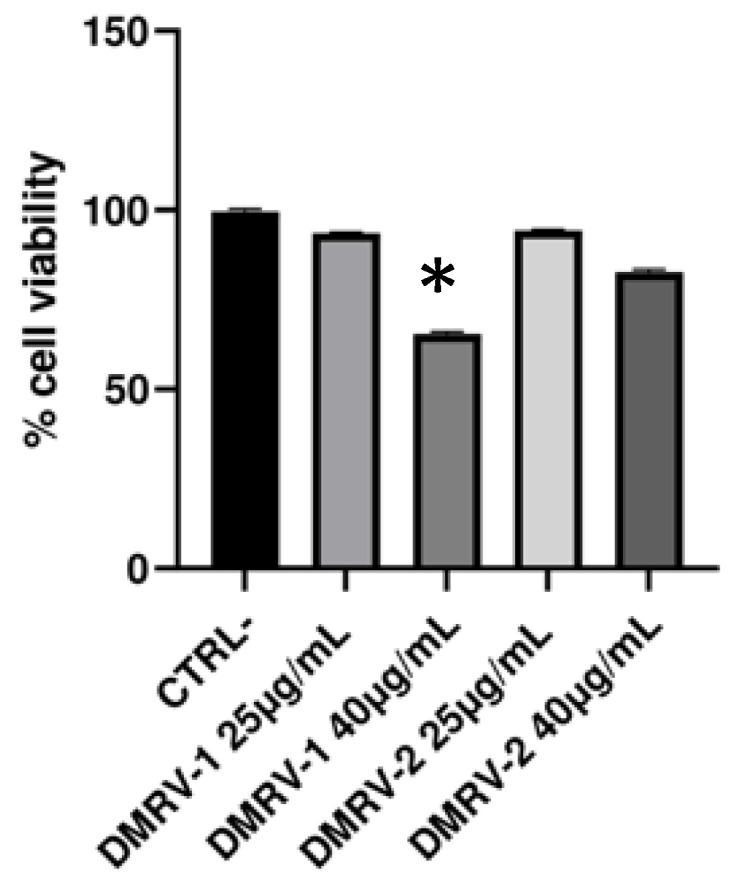
HUVECs viability measured by MTT assay (Mean ± SEM of 6 independent measurements) following different treatments with the two herbal preparations. The cell viability percentage of HUVECs measured after 24 h of incubation with two different concentrations (25 mg/mL and 40 mg/mL) of DMRV-1 (diosmetin 6%, magnolol 10%, *V. vinifera* seeds extract 48%, *R. aculeatus* root extract 36%) and DMRV-2 (diosmetin 11%, magnolol 9%, *V. vinifera* seeds extract 45%, *R. aculeatus* root extract 35%) is shown. CTRL-represents the negative control, consisting of HUVECs incubated with the cell medium for 24 h. * *p* < 0.05 vs. CTRL-.

**Figure 4 plants-12-01051-f004:**
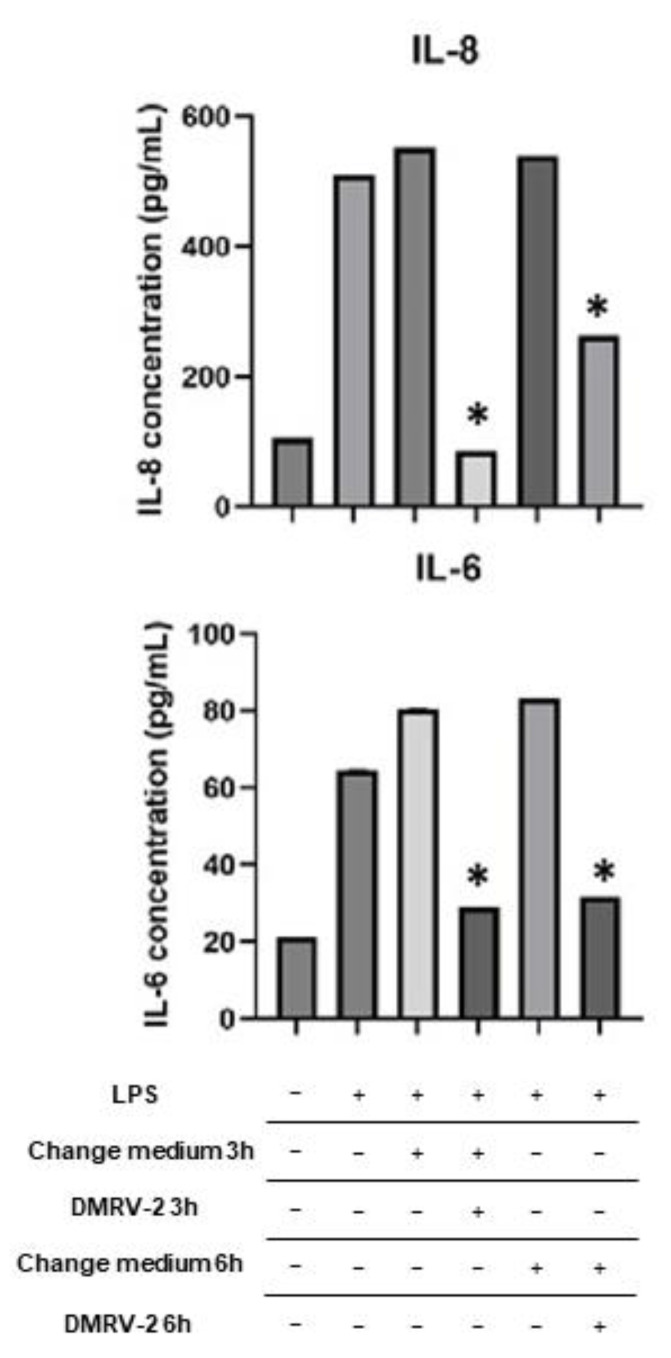
Analysis of IL-6 and IL-8 secretion in HUVECs. HUVECs were stimulated with 0.1 µg/mL LPS for 4 h and then incubated with DMVR-2 or underwent change of the medium. ELISA were performed to measure the concentration of IL-6 and IL-8 released after LPS stimulation, and after 3 h or 6 h of the different incubations. * *p* < 0.05 vs. respective controls.

**Figure 5 plants-12-01051-f005:**
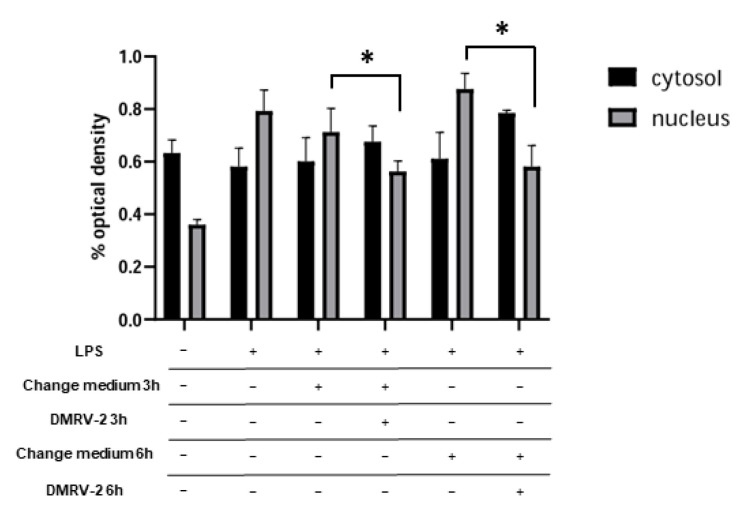
NF-κB translocation analysis. HUVECs were stimulated with 0.1 µg/mL LPS for 4 h and then incubated with DMVR-2 or underwent change of the medium. A WB-based measurement of the NF-κB concentration in the HUVEC cytosol or nucleus following the different treatments was performed. The results of densitometric analysis of the resulting bands, normalized towards the suitable control (nucleolin for nucleus and GAPDH for cytosol) are reported. * *p* < 0.05 vs. respective controls.

**Figure 6 plants-12-01051-f006:**
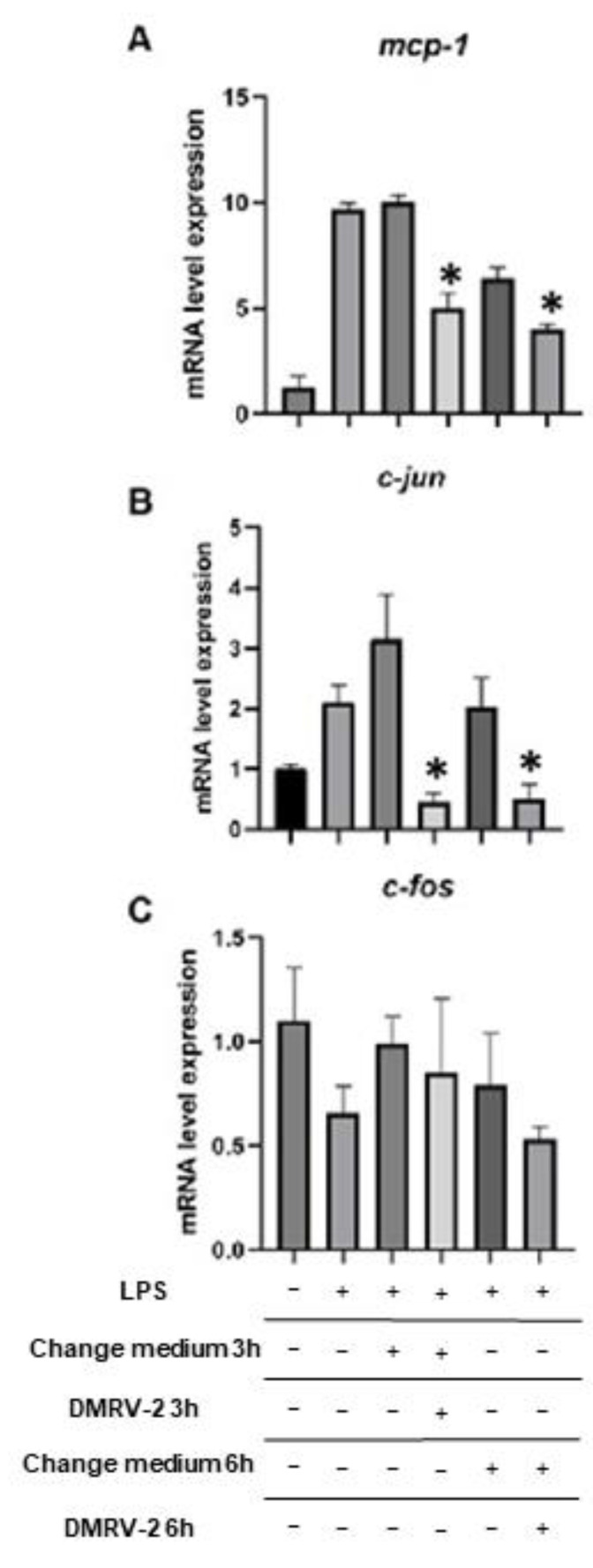
RT-qPCR-measured levels of MCP-1 (**A**), c-Jun (**B**) and c-Fos (**C**) mRNAs in HUVECs following different treatments. HUVECs were stimulated with 0.1 µg/mL LPS for 4 h and then incubated with DMVR-2 or underwent change of the medium. RT-qPCR analyses were performed after LPS stimulation, and after 3 h or 6 h of different incubations. * *p* < 0.05 vs. respective controls.

## Data Availability

Not applicable.

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
