# Peer review of "Design of an Herbal Preparation Composed by a Combination of Ruscus aculeatus L. and Vitis vinifera L. Extracts, Magnolol and Diosmetin to Address Chronic Venous Diseases through an Anti-Inflammatory Effect and AP-1 Modulation"

_plants, 2023, doi:10.3390/plants12051051_

Round 1

Reviewer 1 Report

Manuscript Id: Plant-2111122

 The manuscript reported Raffaella Nocera et al, “A mixture of Ruscus aculeatus L. and Vitis vinifera L. extracts, 2 magnolol and diosmetin to address chronic venous diseases 3 through an anti-inflammatory effect and AP-1 modulationhas merit to be published because of possessing strong novelty. However, I have noted some problems while reviewing this study.

1.      The author needs to re-write the abstract as the flow is well managed.

2.      It would be better to search more literature and add in introduction part.

3.      The conclusion is quite as it needs some main findings.

4.      The resolution of all figures is of very bad quality.

5.      Discussion section should be re written because there are repeated results.

6.      The results need rigid validation especially of AP-1.

7.      The main issue of this paper is the lack of clear language. I strongly suggest to authors to read carefully the paper and to re-write the introduction and also several other sentences that are not clear through all the text. I also suggest to ask the help of a mother tongue in order to have a clearer text.

8.      The author’s needs to on conclusion because it’s too long.

9.      Some references should be included: DOI: 10.26434/chemrxiv-2022-276cm,  "A novel drug candidate for Alzheimer's disease treatment: gx-50 derived from Zanthoxylum bungeanum". Journal of Alzheimer's Disease, 2013. 34(1): p. 203-213; "Destabilization of Alzheimer’s Aβ42 protofibrils with a novel drug candidate wgx-50 by molecular dynamics simulations". The Journal of Physical Chemistry B, 2015. 119(34): p. 11196-11202.

Author Response

Reviewer 1

The manuscript reported Raffaella Nocera et al, “A mixture of Ruscus aculeatus L. and Vitis vinifera L. extracts, 2 magnolol and diosmetin to address chronic venous diseases 3 through an anti-inflammatory effect and AP-1 modulation has merit to be published because of possessing strong novelty. However, I have noted some problems while reviewing this study.

We are very grateful to the reviewer for his suggestions and criticisms, on the basis of which we have substantially modified the manuscript. In our opinion, the new version is much better than the previous one and we hope to have fixed the highlighted problems.

  1. The author needs to re-write the abstract as the flow is well managed.

The abstract has been rewritten according to the reviewer suggestions

  1. It would be better to search more literature and add in introduction part.

The introduction has been changed substantially, taking into account the suggestions of the reviewer. Several references have been added

  1. The conclusion is quite as it needs some main findings.

The conclusions have been rewritten according to the reviewer's suggestions also trying to emphasize the novelty of some of the results achieved.

  1. The resolution of all figures is of very bad quality.

Figures have been prepared with a higher resolution

  1. Discussion section should be re written because there are repeated results.

The result and discussion section has also been rewritten to make it easier to read and understand the results obtained and their importance in our research

  1. The results need rigid validation especially of AP-1.

All reported results are the mean of at least six measurements, derived from two technical analyses of three biological replicates. Validating the real time qtPCR data with other approaches, such as for example the WB analysis on c-Fos, c-Jun and MCP-1, was not possible because the absolute levels of these proteins are quite low and, above all, the variations measured concern their neo-synthesis and not the level they have in the cell after the different treatments. 

  1. The main issue of this paper is the lack of clear language. I strongly suggest to authors to read carefully the paper and to re-write the introduction and also several other sentences that are not clear through all the text. I also suggest to ask the help of a mother tongue in order to have a clearer text.

Large parts of the manuscript have been rewritten

  1. The author’s needs to on conclusion because it’s too long.

The conclusions have been rewritten according to the reviewer's suggestions also trying to emphasize the novelty of some of the results achieved.

  1. Some references should be included: DOI: 10.26434/chemrxiv-2022-276cm, "A novel drug candidate for Alzheimer's disease treatment: gx-50 derived from Zanthoxylum bungeanum". Journal of Alzheimer's Disease, 2013. 34(1): p. 203-213; "Destabilization of Alzheimer’s Aβ42 protofibrils with a novel drug candidate wgx-50 by molecular dynamics simulations". The Journal of Physical Chemistry B, 2015. 119(34): p. 11196-11202.

as suggested by this reviewer, in the new version of the manuscript we have added some new references. however, I do not believe that the papers suggested by the reviewer are consistent with the topics covered in our manuscript and we do not see what useful information they can add to what we have described.

Reviewer 2 Report

The authors propose an interesting study with results obtained using several classical and advanced methods, still, the overall quality of the presentation is too low for a Q1 journal. The manuscript requires drastic modifications of writing style, and text repetitions, errors in results presentations and interpretations reduce the manuscript value. Along with multiple corrections to the text, the authors should insert paragraphs regarding the novelty of the research (introduction section - before the aim and conclusion section), add clear and results-based conclusions.

Some suggestions are listed below:

point 1 - the title should be modified to clearly express the mixture/combination of Ruscus aculeatus L. and Vitis vinifera L. extracts, magnolol and diosmetin

point 2 – the abstract contains little information on the methods and the results, the authors could rephrase the introduction and insert more details regarding the methods and the results

point 3 – the introduction

3.1. it is relatively too long and lines 35-49 should be rephrased. In its current form, the text presented with lines 35-49 is a mixture of information with too much focus on the prevalence data; the authors are recommended to concisely present the pathology definition (according to medical boards/ guidelines), its importance and prevalence/incidence (lines 36-42 and again, but different data with lines 53-54), clinical forms and complications, the polyfactorial etiology and pathogenesis, the treatment protocols/ options.

Ortega MA, Fraile-Martínez O, García-Montero C, Álvarez-Mon MA, Chaowen C, Ruiz-Grande F, Pekarek L, Monserrat J, Asúnsolo A, García-Honduvilla N, Álvarez-Mon M, Bujan J. Understanding Chronic Venous Disease: A Critical Overview of Its Pathophysiology and Medical Management. J Clin Med. 2021 Jul 22;10(15):3239. doi: 10.3390/jcm10153239.

Ligi D, Croce L, Mannello F. Chronic Venous Disorders: The Dangerous, the Good, and the Diverse. Int J Mol Sci. 2018 Aug 28;19(9):2544. doi: 10.3390/ijms19092544.

3.2. the introduction contains at least 3 distinct parts with no links between each other – eg there is no link between lines 35-54 and lines 55-74. The authors should start line 55 with suggested CVD mechanisms etc and include AP-1 pathways

3.3 the authors should remove figure 1 and include a figure with these CVD mechanisms/pathways to better explain the therapeutic effects of the proposed formulation.

point 4 - syntax of several phrases is very difficult to understand, and some of the medical vocabulary formulations are questionable/ not suitable or present a too basic scientific level

eg.

Lines 40-41 – “This pathology has several clinical indications from spider veins, through varicose veins, to venous ulceration”;

lines 43-45 “The cause of venous diseases is the damage to the valves, leads to blood stagnation in the changed vessels; venous hypertension is then transmitted to the microcirculation and results in capillary changes, which lead to edema, skin 45 damage and to venous ulceration. “

lines 80-84 “This research was aimed to develop an herbal product that acted on CVD by abrogating the main damages that this pathology induces and, at transcriptional level blocking the hyperactivation of AP-1. With this aim, an herbal preparation consisting of Ruscus aculeatus L. (Asparagaceae) roots and Vitis vinifera L. (Vitaceae) seed extract, diosmetin and magnolol was set up.”

In terms of research protocols, this proposed formulation containing natural compounds will be used as a treatment - what type of treatment? for what form of the pathology? 

Lines 122-128 “Preliminarily, the possible cytotoxicity towards human endothelial of magnolol, diosmetin, V. vinifera and R. aculeatus extracts, and the herbal preparations cells was evaluated…128” Words repetitions and details of the protocols should be avoided. Most probably the authors intended to write:

The potential in vitro cytotoxicity of herbal preparations constituents (magnolol, diosmetin, V. vinifera and R. aculeatus extracts) was evaluated on human endothelial cells (HUVECs) using the MTT assay. This initial step allowed ..

Point 5. Results section

5.1. Figure 2 legend should be improved - …different amounts ? or (concentrations?) of magnolol, diosmetin, R. aculeatus and V. vinifera extracts; (n=6) should be removed, since the graphs present the average value of the viability % and not 6 distinct values for the same parameters; the authors should add “Values represent the mean ± standard deviation of .. independent measurements.”Add data regarding the acronyms from the graphs

5.2. The graph should be corrected as well – eliminate blank/ctrl column and add controls  - negative (cells without treatment -  most probably is the ctrl column) and positive controls (cells treated with cytotoxic agents)

5.3. More repetitions follow with lines 137-141, the authors should avoid repeating the same text regarding the choice of the preparations and add one concise paragraph about the tested products

5.4. The authors should add details on the results – This description from line 143 “A slight cytotoxicity” is not sufficient

Present and compare the obtained cells viability % (statistical differences! and dose-dependent effect are obvious from the graphs – figure 2)

point 6 - Italic style for Latin names, in vitro etc

point 7 Figure 3 – the same corrections as suggested for Figure 2, as for the legend Figure 3. HUVECs viability after 24 h of incubation with DMRV-1 (2.5 - 4.0 µg/mL of 148 Magnolol) and DMRV-2 (2.5 - 4.0 µg/mL MGL) was assessed by MTT assay (n = 6). It is not an interval, there are two distinct concentrations!

Point 8 – no discussion text for the cytotoxicity of individual constituents – the authors should add literature data and in case there are no references regarding similar studies, this aspect should be underlined as part of the study novelty.

Point 9 sections 3.3. and 3.4 

- All the figures require corrections of the legends

- the same mixture of information - methods, discussion etc, the authors should clearly present the obtained results and relate them to the aim of the study

Point 10. Statistical analysis results should be inserted in the text (p </>)

Point 11  the cell line was obtained from… (research ethics), the viability % was obtained using what formula …

Point 12. The conclusion section should be rephrased by an experienced author to avoid word repetitions, discussions and statements that are not supported by this study results!

– lines 312-320, 324-326 etc do not represent the conclusions of the study;

Line 322 ”…. reduce the severity of symptoms related to CVD” this statement is not supported by the presented results, what symptoms are less severe? Symptoms are not mechanisms or pathways!

Lines 324-326 should be deleted, this is not scientifical style, these are not conclusions “To this aim, we used a mixture of extracts whose anti-inflammatory activity was 324 well documented [as examples see 19,24] and magnolol, a plant metabolite supposed to 325 inhibit AP-1 activity [17]. “

Lines 327-347 should be rephrased to avoid repetitions and errors and moved as discussions

Author Response

Reviewer 2

The authors propose an interesting study with results obtained using several classical and advanced methods, still, the overall quality of the presentation is too low for a Q1 journal. The manuscript requires drastic modifications of writing style, and text repetitions, errors in results presentations and interpretations reduce the manuscript value. Along with multiple corrections to the text, the authors should insert paragraphs regarding the novelty of the research (introduction section - before the aim and conclusion section), add clear and results-based conclusions.

We are very grateful to the reviewer for his suggestions and criticisms, on the basis of which we have substantially modified the manuscript. In our opinion, the new version is much better than the previous one and we hope to have fixed the highlighted problems.

We have almost completely rewritten the introduction trying to better explain the objective and the novelty of our research and we have added further data, relating to the ability of our herbal preparation to block the activation of NF-kB in HUVECs after inflammation.

Some suggestions are listed below:

point 1 - the title should be modified to clearly express the mixture/combination of Ruscus aculeatus L. and Vitis vinifera L. extracts, magnolol and diosmetin

We changed the title to make it clearer

point 2 – the abstract contains little information on the methods and the results, the authors could rephrase the introduction and insert more details regarding the methods and the results

The abstract has been rewritten according to the reviewer suggestions

point 3 – the introduction

3.1. it is relatively too long and lines 35-49 should be rephrased. In its current form, the text presented with lines 35-49 is a mixture of information with too much focus on the prevalence data; the authors are recommended to concisely present the pathology definition (according to medical boards/ guidelines), its importance and prevalence/incidence (lines 36-42 and again, but different data with lines 53-54), clinical forms and complications, the polyfactorial etiology and pathogenesis, the treatment protocols/ options.

Ortega MA, Fraile-Martínez O, García-Montero C, Álvarez-Mon MA, Chaowen C, Ruiz-Grande F, Pekarek L, Monserrat J, Asúnsolo A, García-Honduvilla N, Álvarez-Mon M, Bujan J. Understanding Chronic Venous Disease: A Critical Overview of Its Pathophysiology and Medical Management. J Clin Med. 2021 Jul 22;10(15):3239. doi: 10.3390/jcm10153239.

Ligi D, Croce L, Mannello F. Chronic Venous Disorders: The Dangerous, the Good, and the Diverse. Int J Mol Sci. 2018 Aug 28;19(9):2544. doi: 10.3390/ijms19092544.

3.2. the introduction contains at least 3 distinct parts with no links between each other – eg there is no link between lines 35-54 and lines 55-74. The authors should start line 55 with suggested CVD mechanisms etc and include AP-1 pathways

3.3 the authors should remove figure 1 and include a figure with these CVD mechanisms/pathways to better explain the therapeutic effects of the proposed formulation.

point 4 - syntax of several phrases is very difficult to understand, and some of the medical vocabulary formulations are questionable/ not suitable or present a too basic scientific level

eg.

Lines 40-41 – “This pathology has several clinical indications from spider veins, through varicose veins, to venous ulceration”;

lines 43-45 “The cause of venous diseases is the damage to the valves, leads to blood stagnation in the changed vessels; venous hypertension is then transmitted to the microcirculation and results in capillary changes, which lead to edema, skin 45 damage and to venous ulceration. “

lines 80-84 “This research was aimed to develop an herbal product that acted on CVD by abrogating the main damages that this pathology induces and, at transcriptional level blocking the hyperactivation of AP-1. With this aim, an herbal preparation consisting of Ruscus aculeatus L. (Asparagaceae) roots and Vitis vinifera L. (Vitaceae) seed extract, diosmetin and magnolol was set up.”

The introduction has been changed substantially, taking into account the suggestions of the reviewer. We have described the pathology in a less vague way avoiding repetitions and adding information regarding the pathogenesis and etiology. Figure 1 has been replaced

In terms of research protocols, this proposed formulation containing natural compounds will be used as a treatment - what type of treatment? for what form of the pathology? 

In accordance with the reviewer's criticisms, we have eliminated the sentences concerning the therapeutic use of our preparation since, as he rightly pointed out, our data provided information relating to an action at the biochemical level. We therefore proposed, in the conclusions, the possibility of carrying out further studies to understand the effect of this preparation on subjects affected by CVD.

Lines 122-128 “Preliminarily, the possible cytotoxicity towards human endothelial of magnolol, diosmetin, V. vinifera and R. aculeatus extracts, and the herbal preparations cells was evaluated…128” Words repetitions and details of the protocols should be avoided. Most probably the authors intended to write:

The potential in vitro cytotoxicity of herbal preparations constituents (magnolol, diosmetin, V. vinifera and R. aculeatus extracts) was evaluated on human endothelial cells (HUVECs) using the MTT assay. This initial step allowed ..

These sentences were changed as suggested

Point 5. Results section

5.1. Figure 2 legend should be improved - …different amounts ? or (concentrations?) of magnolol, diosmetin, R. aculeatus and V. vinifera extracts; (n=6) should be removed, since the graphs present the average value of the viability % and not 6 distinct values for the same parameters; the authors should add “Values represent the mean ± standard deviation of .. independent measurements.”Add data regarding the acronyms from the graphs

5.2. The graph should be corrected as well – eliminate blank/ctrl column and add controls  - negative (cells without treatment -  most probably is the ctrl column) and positive controls (cells treated with cytotoxic agents)

The graphs were changed as suggested and the legends of all the figures were improved by adding more information

5.3. More repetitions follow with lines 137-141, the authors should avoid repeating the same text regarding the choice of the preparations and add one concise paragraph about the tested products

Large parts of the manuscript have been rewritten to avoid redundancies

5.4. The authors should add details on the results – This description from line 143 “A slight cytotoxicity” is not sufficient

Present and compare the obtained cells viability % (statistical differences! and dose-dependent effect are obvious from the graphs – figure 2)

point 6 - Italic style for Latin names, in vitro etc

point 7 Figure 3 – the same corrections as suggested for Figure 2, as for the legend Figure 3. HUVECs viability after 24 h of incubation with DMRV-1 (2.5 - 4.0 µg/mL of 148 Magnolol) and DMRV-2 (2.5 - 4.0 µg/mL MGL) was assessed by MTT assay (n = 6). It is not an interval, there are two distinct concentrations!

Point 8 – no discussion text for the cytotoxicity of individual constituents – the authors should add literature data and in case there are no references regarding similar studies, this aspect should be underlined as part of the study novelty.

A paragraph has been added that discusses the cell viability data reported in figures 2 and 3, also comparing our data with those possibly present in the literature

Point 9 sections 3.3. and 3.4 

- All the figures require corrections of the legends

The legends of all the figures were improved by adding more information

- the same mixture of information - methods, discussion etc, the authors should clearly present the obtained results and relate them to the aim of the study

The results have also been rewritten to make it easier to read and understand the results obtained and their importance in our research

Point 10. Statistical analysis results should be inserted in the text (p </>)

Statistical analysis results have be inserted in the text

Point 11  the cell line was obtained from… (research ethics), the viability % was obtained using what formula …

Information concerning HUVECs and the formula used to calculate cell viability % were added

Point 12. The conclusion section should be rephrased by an experienced author to avoid word repetitions, discussions and statements that are not supported by this study results!

– lines 312-320, 324-326 etc do not represent the conclusions of the study;

Line 322 ”…. reduce the severity of symptoms related to CVD” this statement is not supported by the presented results, what symptoms are less severe? Symptoms are not mechanisms or pathways!

Lines 324-326 should be deleted, this is not scientifical style, these are not conclusions “To this aim, we used a mixture of extracts whose anti-inflammatory activity was 324 well documented [as examples see 19,24] and magnolol, a plant metabolite supposed to 325 inhibit AP-1 activity [17]. “

Lines 327-347 should be rephrased to avoid repetitions and errors and moved as discussions

The conclusions have been rewritten according to the reviewer's suggestions also trying to emphasize the novelty of some of the results achieved and eliminating the reference to the treatment of patients

Reviewer 3 Report

Plants-2111122

Line 28: only  cited " named DMRV-2 "; However, in line 104 cited named DMRV-1 and 2  please explain

Keyword: change herbal preparation by the name of the medicinal plants and magnolol and diosmetin

Explain is inflammatory transcription fractor AP-1  or which transcriptional factors were used

Explain MCP-1

Is venous disease or Chronic venous disease

Lines 83 and 84  Letter "L" is not latin 

Line 85 "bark of " is bark of

 Line 180 change IL-1, CD40 as IL-1 and CD40

3.1 section include chromatogram od each extract 

line 228 include voucher of R. aculeatus and where was deposited

2.5  section which drug control used  please include

Author Response

Reviewer 3

We are very grateful to the reviewer for his suggestions and criticisms, on the basis of which we have substantially modified the manuscript. In our opinion, the new version is much better than the previous one and we hope to have fixed the highlighted problems

Line 28: only cited " named DMRV-2 "; However, in line 104 cited named DMRV-1 and 2 please explain

In the new version we have better explained that DMRV-2 was the most promising of the herbal preparation set up.

Keyword: change herbal preparation by the name of the medicinal plants and magnolol and diosmetin

We have changed the keywords as suggested

Explain is inflammatory transcription fractor AP-1 or which transcriptional factors were used

We have rewritten large parts of the manuscript to make the text clearer and eliminate all ambiguous sentences

Explain MCP-1

MCP-1 function was better explained

Is venous disease or Chronic venous disease

We have rewritten large parts of the manuscript to make the text clearer and eliminate all ambiguous sentences

Lines 83 and 84  Letter "L" is not latin 

Line 85 "bark of " is bark of

 Line 180 change IL-1, CD40 as IL-1 and CD40

We have corrected several typos

3.1 section include chromatogram of each extract 

LC-MS chromatograms of the two extracts were inserted as supplementary material (figure S1)

line 228 include voucher of R. aculeatus and where was deposited

This information was inserted in the material and methods section

2.5  section which drug control used  please include

The purpose of the cell viability assays was mainly to define the range of concentrations of the different components that would not interfere with the subsequent cellular analyses. Therefore, in accordance with several other papers (see for example D. Gutiérrez-Praena et al.  Toxicology in Vitro 25 (2011) 1883–1888; Jin X et al. Int J Mol Med. 48 (2021); Zhang X et al Mol Med Rep. 19 (2019):85-92 and many others), we have used only a negative control, which consisted of HUVECs incubated for 24 h with the culture medium alone.

Reviewer 4 Report

The manuscript: "A mixture of Ruscus aculeatus L. and Vitis vinifera L. extracts, magnolol, and diosmetin to address chronic venous diseases through an anti-inflammatory effect and AP-1 modulation" described important information in relation to chronic venous disease. The main contribution of this research is the description of the effects of different components of herbal products of Rubiscus and Vitis on AP-1.

- A few observation points are pointed out in lines 131-136: too large, please review it. 

- Please add in lines 155-157 what could be this work's main subject and hypothesis.

After these suggestions, the manuscript is suitable for publication.

Author Response

the authors thank the reviewer for his advice.

the manuscript has been modified as suggested (see lines 89-90 and 110-116 of this version of the manuscript), also trying to better clarify the purpose of the research.

Round 2

Reviewer 2 Report

The authors performed certain modifications to improve the quality of their manuscript, but the current form is not to be published yet.

I advise the authors to be aware of the low quality of the presentation, the lack of accuracy of their results presentation (eg the cytotoxicity assay) and also of their low ability to underline the novelty of the study.  

Although the authors consider their modifications as substantial and the reviewer evaluation as criticisms, the aim is to publish in a Q1 journal and the following aspects are listed below:

point 1 – Modify the abstract

-        the abstract contains unnecessary details (lines 16-21) and does not clearly present the methods and the results

-   repetition of the word promising is not sufficient to present the results 

-        English language requires corrections (syntax and misspelling)

point 2 – Modify the introduction

The introduction remains a mixture of information, the authors are not able to clearly describe

- CVD pathogenesis

- lines 38-41 are sentences with little scientifical substances and vocabulary, while Plants is a Q1 journal

- Figure 1 presents other keys words/modifications than the listed lines 

- the actions of the proposed mixture refer to the inflammatory process, still the authors consider it innovative compared to the previous therapies

- although the aspect of the therapeutical approach was indicated in the previous review report, the authors do not present the intended protocol for the future formulation 

Point 3 Section 2, add results description for sections 2.1 and 2.2 and correct all the errors. Add pertinent discussions for these sections as well

2.1 - the authors do not describe the LC-MS/MS-based quantification analysis, they do not have discussions ? 

2.2- the acronym HUVEC  includes cells? do the authors need to repeat the word cells?

- the authors do not describe the obtained results - viability %, no statistical analysis as indicated with the first review report !

- what is the base to measure half maximal inhibitory concentrations (IC50) of 7.2 μg/mL, 6.2 μg/mL, 42 μg/mL and 58 μg/mL ?

- did the authors have a positive control for the cytotoxicity?  

- did the authors include ethanol as a control for the ethanolic extracts?

- the legend English language should be corrected

- discussion are absent for section 2.1 and not sufficient and clear for section 2.2

Point 4 Conclusion section

the current form is a subjective discussion, with several words and formulations that have no place in a scientifical paper!

[as examples see 20,25,26], Obviously etc

Author Response

We thank the reviewer for the kind suggestions and have modified the manuscript accordingly.

 Specifically:

 point 1 – Modify the abstract

-        the abstract contains unnecessary details (lines 16-21) and does not clearly present the methods and the results

-   repetition of the word promising is not sufficient to present the results 

-        English language requires corrections (syntax and misspelling)

The abstract was modified, in order to present more clearly the methods and the results. English language was corrected

point 2 – Modify the introduction

The introduction remains a mixture of information, the authors are not able to clearly describe

- CVD pathogenesis

- lines 38-41 are sentences with little scientifical substances and vocabulary, while Plants is a Q1 journal

- Figure 1 presents other keys words/modifications than the listed lines 

Introduction was further modified to describe more clearly some aspects of CVD pathogenesis; figure 1 was modified as well.

- the actions of the proposed mixture refer to the inflammatory process, still the authors consider it innovative compared to the previous therapies

The novelty of the preparation consists of the presence of substances able to act simultaneously on different pathways with different molecular mechanisms. However, following the reviewer's suggestion, we have eliminated the word "innovative" referring to the preparation.

- although the aspect of the therapeutical approach was indicated in the previous review report, the authors do not present the intended protocol for the future formulation 

In his previous report, the reviewer rightly stated that our results did not allow us to hypothesize an effect of our preparation on CVD symptoms (point 12 of the previous report). Accordingly, we have redesigned the manuscript, focusing on the study of the effects of DMRV-2 on the mechanisms and pathways of inflammation in HUVECs. Therefore, considerations regarding the protocol for the future formulation are outside the scope of the manuscript in its present form.

Point 3 Section 2, add results description for sections 2.1 and 2.2 and correct all the errors. Add pertinent discussions for these sections as well

2.1 - the authors do not describe the LC-MS/MS-based quantification analysis, they do not have discussions ? 

LC-MS/MS based quantitation of ruscogenin and procyanidin B1 was performed in order to provide a sort quality control of the extracts. Therefore, data were not discussed. However, we added a short comment and coherent references

2.2- the acronym HUVEC  includes cells? do the authors need to repeat the word cells?

We apologize for the inclusion of the word “cells” after HUVEC. We have eliminated it.

- the authors do not describe the obtained results - viability %, no statistical analysis as indicated with the first review report !

- what is the base to measure half maximal inhibitory concentrations (IC50) of 7.2 μg/mL, 6.2 μg/mL, 42 μg/mL and 58 μg/mL ?

- did the authors have a positive control for the cytotoxicity?  

The purpose of carrying out the cell viability tests was to define the range of concentrations of the different components of the herbal preparations that would not have affected the subsequent cellular assays. We used the data obtained from the MTT assay to calculate the amount of each substance that, after a 24 h treatment, reduced the viability of HUVECs by 50% (IC50), in order to be sure to use lower amounts in subsequent experiments. However, in the present version we included statistical analysis. Moreover, in accordance with several other papers (eg. D. Gutiérrez-Praena et al.  Toxicology in Vitro 25 (2011) 1883–1888; Jin X et al. Int J Mol Med. 48 (2021); Zhang X et al Mol Med Rep. 19 (2019):85-92 and many others), we have used only a negative control.

- did the authors include ethanol as a control for the ethanolic extracts?

As reported in 3.2, the extracts underwent lyophilisation. Therefore, solvents (water and ethanol) used to perform the extraction were completely removed.

- the legend English language should be corrected

Legends were corrected

- discussion are absent for section 2.1 and not sufficient and clear for section 2.2

As assessed, sections 2.1 and 2.2 describe the results of preliminary experiments performed to verify the overall quality of the extracts and to choose the quantity of each component that could be used to possibly maintain the desired activity while minimizing the risk of cellular toxicity. Therefore, the results obtained were almost not discussed in the text. However, in this new version some comments were added.

Point 4 Conclusion section

the current form is a subjective discussion, with several words and formulations that have no place in a scientifical paper!

[as examples see 20,25,26], Obviously etc

We have eliminated the “words and formulations” that the reviewer claims have no place in a scientific paper.

Round 3

Reviewer 2 Report

The authors failed to modify the manuscript according to the detailed recommendations.

In fact, most of these recommendations are disregarded or even criticized by the authors. The authors consider that several sections should not be improved since these sections refer to preliminary steps. Even if these sections refer to preliminary steps, all the methods and the presentation of the results should be accurate.

All the suggestions were detailed, pertinent and well-intended.

The authors manifested a lack of interest to improve the manuscript’s quality and a lack of respect toward any other published papers that present accurate information.

The authors are not able to clearly present the relevance of their results and do not understand the steps required to propose a herbal-based product in therapy

To propose an herbal mixture to replace classical treatment without the basic knowledge and to comment on the reviewer’s comments regarding:

- the herbal chemical characterization? Product characterization and standardization are a must, but the authors are not able to understand this aspect. “sort quality control of the extracts” and short comment” are not sufficient. It is not accurate to evaluate a herbal product and propose it for therapy based on its biological properties without chemical characterization and standardization of the herbal product.

- cytotoxicity level and results presentation and interpretation

- conclusion section is not suitable, the authors disregarded the reviewer’s observations

 point 1 – Modify the abstract

-        the abstract contains unnecessary details (lines 16-21) and does not clearly present the methods and the results

-   repetition of the word promising is not sufficient to present the results 

Authors response: The abstract was modified, in order to present more clearly the methods and the results. English language was corrected

Reviewer Response: The abstract was modified by adding word formulations that are not suitable for a Q1 journal and details that do not point out the results or the conclusions

- the abstract still contains unnecessary details (lines 16-21), with the first introductory phrase having no connection with the aim of the study and the second phrase being too detailed and vague

- word formulations that are not suitable for a Q1 journal: “Many therapies have been proposed to deal with CVD, but unfortunately the symptoms recur with increasing frequency and intensity as soon as treatments are stopped.” The authors should rephrase to clearly link the reoccurrence of the pathology to the mechanism they studied (if such connections are valid)

- the rest of the text – lines 22 -36 – does not follow the basic structure of an abstract, I recommend the author to concisely describe the methodology, to detail the results and to add conclusions.

point 2 – Modify the introduction

The introduction remains a mixture of information, the authors are not able to clearly describe

- CVD pathogenesis

- lines 38-41 are sentences with little scientifical substances and vocabulary, while Plants is a Q1 journal

- Figure 1 presents other keys words/modifications than the listed lines 

Introduction was further modified to describe more clearly some aspects of CVD pathogenesis; figure 1 was modified as well.

Reviewer Response: not performed

- the actions of the proposed mixture refer to the inflammatory process, still the authors consider it innovative compared to the previous therapies

The novelty of the preparation consists of the presence of substances able to act simultaneously on different pathways with different molecular mechanisms. However, following the reviewer's suggestion, we have eliminated the word "innovative" referring to the preparation.

Reviewer Response: any scientifical article should present the novelty, the authors should be able to name it in the manuscript text

- although the aspect of the therapeutical approach was indicated in the previous review report, the authors do not present the intended protocol for the future formulation 

In his previous report, the reviewer rightly stated that our results did not allow us to hypothesize an effect of our preparation on CVD symptoms (point 12 of the previous report). Accordingly, we have redesigned the manuscript, focusing on the study of the effects of DMRV-2 on the mechanisms and pathways of inflammation in HUVECs. Therefore, considerations regarding the protocol for the future formulation are outside the scope of the manuscript in its present form.

Reviewer Response: The authors are not able to clearly present the relevance of their results and do not understand the steps required to propose a herbal based product in therapy

Point 3 Section 2, add results description for sections 2.1 and 2.2 and correct all the errors. Add pertinent discussions for these sections as well

2.1 - the authors do not describe the LC-MS/MS-based quantification analysis, they do not have discussions ? 

LC-MS/MS based quantitation of ruscogenin and procyanidin B1 was performed in order to provide a sort quality control of the extracts. Therefore, data were not discussed. However, we added a short comment and coherent references

Reviewer Response: “sort quality control of the extracts” and short comment” are not sufficient. It is not accurate to evaluate a herbal product and propose it for therapy based on its biological properties without chemical characterization and standardization of the herbal product.

- the authors do not describe the obtained results - viability %, no statistical analysis as indicated with the first review report !

- what is the base to measure half maximal inhibitory concentrations (IC50) of 7.2 μg/mL, 6.2 μg/mL, 42 μg/mL and 58 μg/mL ?

- did the authors have a positive control for the cytotoxicity?  

The purpose of carrying out the cell viability tests was to define the range of concentrations of the different components of the herbal preparations that would not have affected the subsequent cellular assays. We used the data obtained from the MTT assay to calculate the amount of each substance that, after a 24 h treatment, reduced the viability of HUVECs by 50% (IC50), in order to be sure to use lower amounts in subsequent experiments. However, in the present version we included statistical analysis. Moreover, in accordance with several other papers (eg. D. Gutiérrez-Praena et al.  Toxicology in Vitro 25 (2011) 1883–1888; Jin X et al. Int J Mol Med. 48 (2021); Zhang X et al Mol Med Rep. 19 (2019):85-92 and many others), we have used only a negative control.

- discussion are absent for section 2.1 and not sufficient and clear for section 2.2

As assessed, sections 2.1 and 2.2 describe the results of preliminary experiments performed to verify the overall quality of the extracts and to choose the quantity of each component that could be used to possibly maintain the desired activity while minimizing the risk of cellular toxicity. Therefore, the results obtained were almost not discussed in the text. However, in this new version some comments were added.

Point 4 Conclusion section

the current form is a subjective discussion, with several words and formulations that have no place in a scientifical paper!

[as examples see 20,25,26], Obviously etc

We have eliminated the “words and formulations” that the reviewer claims have no place in a scientific paper.

Reviewer response: the current form of the conclusion section is not suitable; the authors did not perform the suggested modifications

Author Response

We regret that the reviewer did not consider our changes adequate and sufficient to make our manuscript eligible for publication in Plants. We are very grateful for the advice he/she and other reviewers have given us and we believe these have helped improve our product. But now, we do not understand these new criticisms that once again challenge the reporting style of the data, and not our results.